# CTHRC1 Induces Pancreatic Stellate Cells (PSCs) into Myofibroblast-like Cancer-Associated Fibroblasts (myCAFs)

**DOI:** 10.3390/cancers15133370

**Published:** 2023-06-27

**Authors:** Min Kyung Kang, Fen Jiang, Ye Ji Kim, Kyoungjin Ryu, Atsushi Masamune, Shin Hamada, Yun-Yong Park, Sang Seok Koh

**Affiliations:** 1Department of Biomedical Sciences, Dong-A University, Busan 49315, Republic of Korea; 2Prestige BioPharma IDC Co., Ltd., Busan 46726, Republic of Korea; 3Division of Gastroenterology, Tohoku University Graduate School of Medicine, Sendai 980-8574, Japan

**Keywords:** cancer-associated fibroblast, cell differentiation, CTHRC1, extracellular matrix, microenvironment myofibroblast, neoplasm metastasis, pancreatic cancer, pancreatic stellate cell, periostin

## Abstract

**Simple Summary:**

CTHRC1 is a protein with pro-tumoral effects that is highly expressed in tumors. We found that cancer stroma is the major source of CTHRC1 secretion in pancreatic cancer, and CTHRC1 regulates cancer extracellular matrix (ECM) remodeling by activating pancreatic stellate cells (PSCs), a main component of pancreatic cancer stroma. Our results show that CTHRC1-activated PSCs differentiated into myofibroblast-like CAFs (myCAFs). Moreover, the pro-tumoral effects manifested by CTHRC1 were more potent through PSC activation than via autocrine action. Periostin, a stroma-specific molecule, demonstrated an essential role in the CTHRC1–PSCs–cancer metastasis axis. Blocking CTHRC1 reversed PSC activation in vitro and showed significant anti-tumoral effects in animal models. However, long-term CTHRC1 treatment likely caused a shift from myCAFs into inflammatory CAFs. This study elucidates PSC activation as the central mechanism underlying the pro-tumoral effects of CTHRC1 and suggests that future studies on CTHRC1 as a potential therapeutic target for pancreatic cancer are warranted.

**Abstract:**

**[BACKGROUND]** Collagen triple helix repeat containing-1 (CTHRC1) is a secreted protein that contributes to the progression of various cancers, including pancreatic cancer. The higher expression of CTHRC1 in tumor tissues is associated with poorer survival outcomes. However, its specific roles in tumor extracellular matrix (ECM) remodeling remain unclear. Our study aims to investigate the influences of CTHRC1 on pancreatic stellate cells (PSCs), a main source of ECM production in pancreatic cancer. **[METHODS AND RESULTS]** The analyses of the publicly available pancreatic cancer patient data revealed that CTHRC1 is mainly expressed in cancer stroma and highly correlated with ECM-related genes. An in vitro study showed that more than 40% of these genes can be upregulated by CTHRC1. CTHRC1 specifically activated PSC into myofibroblast-like cancer-associated fibroblasts (myCAFs), which are characterized by a significantly upregulated POSTN gene expression. Periostin (coded by the POSTN gene) has a central role in the CTHRC1–PSCs–cancer metastasis axis. Furthermore, CTHRC1 promoted pancreatic cancer cell proliferation through PSC activation to a greater extent than via direct stimulation. Proof-of-concept experiments showed that the long-term (4-week) inhibition of CTHRC1 led to significant tumor suppression and ECM reduction, and also resulted in an unexpected shift in the CAF subtype from myCAFs to inflammatory CAFs (iCAFs). **[CONCLUSION]** PSC activation was demonstrated to be the key molecular mechanism responsible for the tumor-promoting effects of CTHRC1, and CTHRC1 has a critical role in CAF subtype differentiation and tumor microenvironment (TME) remodeling. The inhibition of CTHRC1 as a therapeutic strategy for the treatment of pancreatic cancer warrants further investigation.

## 1. Introduction

Pancreatic cancer is one of the most lethal cancers worldwide. Despite ongoing efforts to improve diagnosis and treatment, the median 5-year survival rate remains at only 10% [1]. Pancreatic cancer has a unique tumor microenvironment (TME) characterized by a dense stromal shield that is formed through desmoplasia. Desmoplasia is driven by the activation of pancreatic stellate cells (PSCs), which are located in the pancreatic stroma [2]. While PSCs can display as a protective mechanism, which prevents tumor cells from spreading [3], they are more notoriously known for their pro-tumoral roles, owing to the heterogeneity of the PSC population [4].

Collagen triple helix repeat containing-1 (CTHRC1) is a secreted 26 kDa protein. It was first identified in injured arteries and was found to contribute to vascular remodeling by limiting collagen matrix deposition and promoting cell migration [5]. CTHRC1 overexpression has been recently identified in various solid tumors, including pancreatic, liver, ovarian, breast, and lung cancers [6]. CTHRC1 promotes tumor progression in both autocrine and paracrine manners; it stimulates cancer cell migration/invasion [7] and proliferation [8] directly, and also fuels tumor growth indirectly via tumor angiogenesis [9]. The molecular mechanisms of the autocrine action of CTHRC1 include increasing the adhesion of cancer cells to the extracellular matrix (ECM) through the induction of integrin β1 expression and the activation of focal adhesion kinase [10]. CTHRC1 has been demonstrated to interact with stellate cells in the liver (HSCs) and promote liver fibrosis [11]. However, the interactions between CTHRC1 and PSC and the roles of such paracrine actions of CTHRC1 in the remodeling of the pancreatic cancer TME remain unknown.

This study aims to examine the influence of CTHRC1 on PSCs and to investigate the role of PSCs in the tumor-promoting effect of CTHRC1 at the molecular level. Proof-of-concept studies using two different human pancreatic cancer mouse models are performed to investigate the effects of an anti-CTHRC1 monoclonal antibody (α-CTHRC1) treatment on ECM formation and the differentiation of cancer-associated fibroblasts (CAFs).

## 2. Materials and Methods

### 2.1. Cells

Human pancreatic cancer cell lines MIA PaCa-2, PANC-1, BxPC-3, and CFPAC-1 were purchased from the American Type Culture Collection (ATCC, Manassas, VA, USA). Short tandem repeats (STR) fingerprinting was performed to confirm their identities. The MIA PaCa-2 and PANC-1 cell lines were maintained in Dulbecco’s modified Eagle’s medium (DMEM, Cytiva, Marlborough, MA, USA), the BxPC-3 cell line was maintained in RPMI-1640 (RPMI, Cytiva), and the CFPAC-1 cell line was maintained in Iscove’s modified Dulbecco’s medium (IMDM, Cytiva). The HEK293T cell line was gifted by Prof. Dae-Sik Lim of the Korea Advanced Institute of Science and Technology (KAIST, Daejeon, Republic of Korea). The immortal human PSC line, hPSC21-S/T, was provided by Prof. Tooru Shimosegawa of Tohoku University (Sendai, Japan) [12]. The POSTN-knockout PSC cell line PSC_POSTN^KO^ was generated using a similar method to what has been previously described [13]. Guide RNA sequences targeting POSTN were designed using the CRISPR design website (http://crispr.mit.edu/, accessed on 1 April 2023). The sequences POSTN sgRNA forward and reverse oligonucleotides (5′-CACCGGGTGCCCAAAATCTGTTGAA-3′ and 5′-AAATTCAACAGATTTTGGGCACCC, respectively) were cloned into the LentiCRISPR vector (Addgene, Watertown, MA, USA). After infecting PSC with POSTN^KO^ lentivirus, the knockout of the POSTN gene was confirmed by the quantitative reverse transcription real-time polymerase chain reaction (RT-qPCR) and enzyme-linked immunosorbent assay (ELISA) of the protein expression of Cas9 and periostin. All cells were supplemented with 10% fetal bovine serum (FBS, Cytiva) and 1% penicillin/streptomycin (Gibco, Grand Island, NY, USA), cultured at 37 °C in 5% CO_2,_ and monitored for mycoplasma contamination using a mycoplasma detection kit (CellSafe, Yongin, Republic of Korea).

### 2.2. Reagents

Recombinant CTHRC1-His (noted as CTHRC1) solution was prepared as previously described [7], which was determined to be a mixture of monomer (26 kD), dimer (52 kD), and trimer (78 kD) forms of CTHRC1, and the respective molar percentages of each CTHRC1 form were 25%, 27%, and 48%, respectively, as determined by mass spectrometry. Therefore, the average molecular weight of the CTHRC1 mixture was determined to be 57.98 kD (26 × 25% + 52 × 27% + 78 × 48%). Accordingly, we calculated the molar concentration of 100 ng/mL CTHRC1 to be 1.7 nM. An anti-CTHRC1 humanized monoclonal antibody (noted as α-CTHRC1) was prepared as described in Patent KR102487687B1. Recombinant TGF-β protein was purchased from PeproTech (cat# 100-21, Rocky Hill, NJ, USA). Human IgG1 isotype control (cat# 31154) was obtained from Thermo Fisher Scientific (Waltham, MA, USA). Antibodies against α-SMA (cat# ab5694), FAP (cat# ab53066), and Ki67 (cat# ab15580) were purchased from Abcam (Cambridge, UK). Antibodies against fibronectin (cat# 26836), collagen type I (cat#84336), and periostin (cat# 91771) were obtained from Cell Signaling Technology (Danvers, MA, USA). Antibodies against GAPDH (cat# sc-47724) and anti-rabbit IgG-FITC were purchased from Santa Cruz Biotechnology (Dallas, TX, USA). Anti-mouse IgG-HRP and anti-rabbit IgG-HRP were purchased from Thermo Fisher Scientific. For the migration/invasion assays, Giemsa (cat# 32884, Sigma-Aldrich, St. Louis, MO, USA) was used as a staining solution. For the invasion assay, Matrigel (cat# 354248, Corning, MA, USA) was used for coating the membrane. For protein detection in Western blot analyses, ECL^TM^ reagents (cat# K-12045-D50, Advansta, San Jose, CA, USA) were used. For the immunofluorescence assay, DAPI (cat# D9542) was purchased from Sigma-Aldrich. VECTASHIELD antifade mounting medium (cat# H1900) was purchased from Vector Laboratories Inc. (Newark, CA, USA). For the proliferation assay, the WST-1 reagent (cat# 11644807001, Sigma-Aldrich) was used.

### 2.3. Analyses of mRNA Data Using Public Databases

To compare the gene expression of CTHRC1 between tumor and non-tumor tissue samples as well as pancreatic cancer cells and stromal cells, publicly available datasets were retrieved from the National Center for Biotechnology Information (NCBI) Gene Expression Omnibus (GEO) database (https://www.ncbi.nlm.nih.gov/gds, accessed on 1 April 2023). Only datasets with a minimum of 30 samples of paired tumor/non-tumor or paired cancer/stroma samples were used. If different datasets contained overlapping samples, the samples in only one dataset were used. The CTHRC1 gene expression data of the cancer cell lines obtained from the Cancer Cell Line Encyclopedia (CCLE) [14] were used to verify our experiment results (https://depmap.org/portal/gene/CTHRC1, accessed on 1 April 2023). The Single Cell Portal database was used to retrieve CTHRC1 expression in cell subsets from pancreatic tumor tissue (study identifier: human treatment-naïve PDAC sNuc-seq, https://singlecell.broadinstitute.org/single_cell/study/SCP1089/human-treatment-naive-pdac-snuc-seq, accessed on 1 April 2023). The cBioPortal database [15,16] was used to retrieve human pancreatic cancer mRNA data and survival data (TCGA, PanCancer Atlas, https://www.cbioportal.org/study/summary?id=laml_tcga_pan_can_atlas_2018, accessed on 1 April 2023; and CPTAC, Cell 2021, https://www.cbioportal.org/study/summary?id=paad_cptac_2021, accessed on 1 April 2023). All patient samples with available mRNA expression profiles from the TCGA project were included in the correlation analysis. All data were downloaded and processed using Biometric Research Branch (BRB, National Cancer Institute, Bethesda, MD, USA) array tools or Prism version 9.0 software (GraphPad Software, San Diego, CA, USA).

### 2.4. RNA Isolation and Quantitative Real-Time PCR (RT-qPCR)

Total RNA was extracted using Trizol reagent (FAVORGEN, Wien, Austria) and used for cDNA synthesis. The cDNA was synthesized using PrimeScript^TM^ RT Master Mix (Takara Bio, Kusatsu, Shiga, Japan), according to the manufacturer’s instructions. RT-qPCR analyses were conducted using the AriaMx Real-Time PCR system (Agilent Technologies, Santa Clara, CA, USA). The primer sequences for analyses are listed in Appendix A. Gene expression levels were normalized to those of the POLR2A or GAPDH genes.

### 2.5. Western Blot Analyses

Cell lysates were prepared in RIPA buffer (50 mM Tris-HCl, 150 mM NaCl, 1% sodium deoxycholate, 5 mM EDTA, 30 mM Na_2_HPO_4_, 50 mM NaF, and 1 mM Na_3_OV_4_), and protein concentrations were determined using the Pierce^TM^ BCA Protein Assay Kit (cat# 23225, Thermo Fisher Scientific). Samples containing equal amounts of cell lysate were loaded onto SDS-PAGE gel and transferred to a nitrocellulose membrane. The membranes were blotted with a corresponding primary antibody in blocking buffer (Tris-buffered saline/Tween-20 containing 5% skim milk) and incubated with horseradish peroxidase-conjugated secondary antibodies. The signal of the band was detected by the Azure C300 gel imaging system (Azure Biosystems, Dublin, CA, USA).

### 2.6. Immunofluorescence (IF) Assay

PSCs were seeded on glass coverslips and treated with CTHRC1 (1.7 nM) or TGF-β (2.1 nM). After 24 h, the cells were fixed in 4% paraformaldehyde (#P6148, Sigma), and permeabilized using 0.2% Triton X-100 (cat# 93443, Sigma-Aldrich). The cells were incubated with anti-FAP or anti-α-SMA antibody at 4 °C for 1 h and then washed with PBST (0.05% Tween 20 in PBS), and stained with FITC-conjugated anti-rabbit secondary antibody. The nuclei were stained with DAPI (0.1 μg/mL) at 37 °C for 10 min. Slides were mounted using VECTASHIELD antifade mounting medium and images were acquired with an Eclipse Ni microscope (Nikon, Tokyo, Japan).

### 2.7. Oil Red O Staining

PSCs were seeded in 6-well plates and treated with CTHRC1 (1.7 nM) or TGF-β (2.1 nM). After 24 h, a lipid (Oil Red O) staining kit (cat# K580, BioVision, Waltham, MA, USA) was used to detect lipid droplets in the PSCs. The cells were fixed with formalin and covered by Oil Red O solution for 20 min. The solution was removed and the plates were washed with distilled water. Images were acquired using an optical microscope (Carl Zeiss, Oberkochen, Germany). Oil Red O was extracted using 100% isopropanol for quantification and its absorbance was read at 492 nm.

### 2.8. Migration and Invasion Assay

Two different sets of migration and invasion assays were performed using Transwell plates with a pore size of 8 μm (cat# 3422, Corning, Corning, NY, USA). In the first set, PSCs (5 × 10^4^) were seeded in the upper chambers with CTHRC1 (1.7 nM), TGF-β (2.1 nM), with or without α-CTHRC1 (69.4 nM). In the second set, pancreatic cancer cells (MIA PaCa-2, PANC-1, and BxPC-3 were seeded at concentrations ranging from 3 × 10^4^ to 8 × 10^4^/well) were seeded in the upper chambers with PSC-conditioned media. In the invasion assay, the insert membrane of the upper chamber was coated with Matrigel. The complete growth media for each cell line was placed in the lower chamber to allow the observation of cell migration and invasion. After 24 h, the cells that migrated or invaded the membrane were stained with Giemsa, and the number of cells in the same-sized three random fields of each Transwell plate were counted using ImageJ software (Version 1.53, Bethesda, MD, USA), and the mean values were calculated.

The POSTN-knockout and control PSC (PSC-POSTN^KO^ and PSC-sgNTC)-conditioned media for the treatment of cancer cells for the second set was prepared as follows: The culture media was replaced with serum-free media to exclude the effect of FBS, before CTHRC1 treatment. Additionally, because CTHRC1 was reported to increase the migration and invasion ability of pancreatic cancer cells, magnetic beads were conjugated to CTHRC1 (or BSA) using a magnetic conjugation kit (cat# ab269890, Abcam). PSC-POSTN^KO^ were treated with the magnetic bead-conjugated CTHRC1 (or BSA). CTHRC1 (or BSA) was then removed using a magnetic stand (cat#12321D, Invitrogen, Waltham, MA, USA), and the conditioned media was harvested. The removal of CTHRC1 from the conditioned media was confirmed through CTHRC1 sandwich ELISA (Appendix A).

### 2.9. Sandwich Enzyme-Linked Immunosorbent Assay (ELISA)

Collagen type I, fibronectin, and periostin were measured in PSCs by sandwich ELISA using a human collagen type I ELISA kit (cat# LS-F22003-1, LS Bio, Lynnwood, WA, USA), human fibronectin ELISA kit (cat# DY-1918-05, R&D Systems, Minneapolis, MN, USA), and human periostin ELISA kit (cat# DY3548B, R&D systems), respectively, following the manufacturers’ instructions. For the measurement of CTHRC1 using sandwich ELISA, pancreatic cancer cells and PSCs were seeded at 1 × 10^6^ cells in 100 mm dishes and cultured for 48 h. After cell debris was removed by centrifugation, the culture supernatant was harvested for the detection of CTHRC1. The plates were coated with 5 μg/mL of CR1P (anti-CTHRC1 polyclonal antibody) and incubated at 4 °C overnight before being incubated with the supernatant for 90 min. The plates were then treated with biotinylated 4H5 (anti-CTHRC1 monoclonal antibody) as the detection antibody at 37 °C for 90 min followed by Streptavidin-HRP (1:5000) at 37 °C for 30 min. CTHRC1 protein expression levels were detected by a microplate reader at the wavelengths of 450 nm and 650 nm.

### 2.10. Cell Proliferation

MIA PaCa-2 and BxPC-3 were seeded in 96-well plates at 2 × 10^3^ cells per well. After 24 h, the cells were treated with recombinant CTHRC1 or a conditioned media of PSCs for 48 h. WST-1 solution was added at a volume of 10% of media per well and the absorbance was read by a microplate reader at a wavelength of 450 nm.

### 2.11. siRNA Transfection

Small interfering RNA duplexes targeting CTHRC1 were purchased from Sigma and transfected using Lipofectamine^TM^ 2000 (cat# 11668027, Invitrogen), following the manufacturer’s protocol. The silencing efficiency of siRNA was assessed by RT-qPCR.

### 2.12. Animal Study

Animal studies were approved by the Institutional Animal Care and Use Committee of Dong-A University, Busan, Republic of Korea (DIACUC-22-26, approval date: 12 July 2022).

A BxPC-3 subcutaneous xenograft mouse model was established in 4-week-old female BALB/c nude mice (Hana Biotech, Pyeongtaek, Republic of Korea). Around 1 × 10^6^ cells were mixed with Matrigel in a 1:1 ratio and inoculated subcutaneously into the right flank of each mouse. Mice were divided into three groups (IgG, *n* = 8; anti-CTHRC1 antibody, *n* = 8; and gemcitabine *n* = 8). After 7 days, the average volume of the tumors reached around 50 mm^3^. IgG, anti-CTHRC1 antibody (10 mg/kg), or gemcitabine (5 mg/kg) were injected intravenously, twice a week (3–4 day intervals) for 6 weeks, and the body weight of each mouse was measured twice a week. All mice were sacrificed on day 45. The size of the tumors was measured with calipers.

A CFPAC-1 orthotopic xenograft mouse model was established in 4-week-old male BALB/c nude mice. A total of 1 × 10^6^ cells were injected orthotopically into the pancreas of each mouse. The mice were randomly divided into four groups (PBS, *n* = 15; gemcitabine, *n* = 16; α-CTHRC1, *n* = 16; and gemcitabine + α-CTHRC1, *n* = 17). After 7 days, the mice were treated twice a week (3–4 day intervals) with gemcitabine (1 mg/kg) and/or anti-CTHRC1 antibody (10 mg/kg). The gemcitabine was intraperitoneally injected and the antibody was intravenously injected. All mice were sacrificed on day 33 (after 4 weeks of treatment). Tumors located in the pancreas were isolated and their weight and volume were measured. Tumor volume was calculated with the following formula:Tumor Volume (mm^3^) = Length (mm) × Width (mm) × Height (mm) × 0.52.

### 2.13. Immunohistochemistry Staining

Sirius Red and Masson’s Trichrome were used for collagen staining. Although the dyes have different amino acid-binding preferences for collagen fibers, a previous study showed that there is no difference in collagen staining between the two methods [17]. For Sirius Red staining, formalin-fixed and paraffin-embedded tumor tissue slices were deparaffinized and rehydrated. The sections were stained using a Picro-Sirius Red staining kit (cat# ab150681, Abcam). Tissue sections were incubated with Picro-Sirius Red solution for 60 min and quickly rinsed twice using an acetic acid solution. Slides were then rinsed in absolute alcohol until clear.

For Masson’s Trichrome staining, tumor tissue samples were fixed in formalin and embedded in paraffin. Sections were stained using a trichrome stain kit (cat# ab150686, Abcam). Deparaffinized sections were incubated in preheated (54–64 °C) Bouin’s Fluid for 60 min. The slides were rinsed in tap water until completely clear and then stained with Weigert’s A reagent and Weigert’s B reagent, which were mixed at an equal ratio. The slides were rinsed with tap water for 2 min and Biebrich Scarlet/Acid solution was applied for 15 min, then rinsed with distilled water. The slides were incubated in a phosphomolybdic/phosphotungstic acid solution for 10 min. Without rinsing, Aniline Blue solution was applied to the slides for 5 min, and then rinsed in distilled water. The slides were incubated in 1% acetic acid solution for 5 min and dehydrated quickly in 95% alcohol, and then rinsed twice in absolute alcohol.

For Ki67 immunohistochemistry staining, the slides were prepared at 4 μm thickness. The assay was performed by a fully automated process using the Leica BOND RX^m^ (Leica Biosystems, Richmond, IL, USA) automated stainer. An antibody against Ki67 was used as the primary antibody and BOND Polymer Refined Detection (cat# DS9800, Leica Biosystems) was used for detection. All slides were cleared in xylene and mounted using Leica CV5030 fully automated glass coverslips (Leica Biosystems). All images were captured using the Leica GT450 automated slide scanner (Leica Biosystems). Collagen deposition and Ki67-positive tumor cells were quantified using ImageJ software (Version 1.53, Bethesda, MD, USA).

### 2.14. Statistical Analyses

All quantitative data from multiple measurements were presented as mean ± standard deviation (SD). Prism version 9.0 software (GraphPad Software, (GraphPad Software, San Diego, CA, USA) was used to perform the statistical analyses. For the GEO datasets, paired *t*-tests were used for the data comparisons of the paired samples. For in vitro experiments, the comparisons of results influenced by two different factors (e.g., cell lines and treatments) were performed using two-way analysis of variance (ANOVA) tests. The comparisons of results influenced by one factor (e.g., treatments) were performed using one-way ANOVA tests. Post-hoc analyses were performed using multiple comparison tests recommended by the Prism software. Dose-dependent effects were analyzed using linear regression after the log-transformation of drug dose. Survival data were analyzed using the Logrank test and the survival difference was presented as a hazard ratio (HR) with 95% confidence intervals (95% CI). The low and high expression groups of POSTN and CTHRC1 were divided when the highest statistical difference was reached (with the highest HR and the lowest *p*-value). For group division using a combination of POSTN and CTHRC1 (POSTN^2^ × CTHRC1), more weight was given to POSTN because it showed a higher statistical survival difference than CTHRC1. The results were considered significant when *p* < 0.05.

## 3. Results

### 3.1. CTHRC1 Expression in Pancreatic Tumor Tissue

To compare the gene expression of CTHRC1 between tumor and non-tumor tissue samples as well as pancreatic cancer cells and stromal cells, publicly available datasets were retrieved from the NCBI GEO database (see Materials and Methods, Section 2.3., Analyses of mRNA data using public databases). As a result, four datasets (GSE62452, GSE183795, GSE15471, and GSE16515) met the inclusion criteria and were used for the comparison of CTHRC1 expression between pancreatic tumor and paired adjacent normal tissue samples. Two datasets (GSE93326 and GSE164665) met the inclusion criteria and were used for the comparison of CTHRC1 mRNA expression between paired pancreatic cancer cells (epithelium) and stromal cells (Appendix A).

The RNA-sequencing data retrieved from the datasets showed the mRNA expression of CTHRC1 in pancreatic tumor tissue was significantly higher than in the paired adjacent non-tumor tissue samples (Figure 1A). Furthermore, within the same pancreatic tumor tissue, stromal cells showed a much higher CTHRC1 expression than in epithelium (cancer) cells (Figure 1B). Our CTHRC1 protein expression data from various pancreatic cancer cell lines and PSCs showed that CTHRC1 expression was much higher in PSCs than most pancreatic cancer cell lines (MIA PACA-2, PANC-1, and CFPAC-1), except for BxPC-3, which expressed CTHRC1 at a rate almost twice as high as PSCs and almost 20 times higher than MIA PaCa-2. In addition, our data are in agreement with the corresponding data from the CCLE database (Figure 1C).

To ascertain the expression status of CTHRC1 in different stromal cells, the single-nucleus RNA-seq data of treatment-naïve pancreatic tumor tissue samples were retrieved from the Single-Cell Portal. CTHRC1 was much more extensively expressed by fibroblast cells than cancer cells, immune cells, and endothelial cells, and other stromal cells had a much lower CTHRC1 expression (Figure 1D).

Two programs (TCGA and CPTAC) containing relevant pancreatic cancer survival data and CTHRC1 tumor expression data were retrieved from the cBioPortal (Figure 1E). The data analyses revealed around two times higher risk of death in the CTHRC1_High groups, compared to the CTHRC1_Low groups (HR = 2.2 and 1.8 for the TCGA and CPTAC programs, respectively). The difference was statistically significant in the TCGA program (*p* = 0.007) only, while the difference in the CPTAC program was marginal (*p* = 0.068), likely due to the much smaller sample size in the CPTAC program (TCGA, *n* = 177; and CPTAC, *n* = 91).

### 3.2. CTHRC1 Expression in Pancreatic Tumor Tissue Correlated with the Expression of ECM-Related Genes

ECM-related genes upregulated by CTHRC1 were identified as follows (Figure 2A): Pancreatic tumor mRNA correlation analyses were conducted between CTHRC1 and 20,530 genes, using the mRNA expression of pancreatic cancer tissue samples (*n* = 183) obtained from the TCGA PanCancer Atlas. Of these, 58 genes were identified to be CTHRC1-correlated with coefficient r > 0.8. The function description of the 58 genes showed that 35 (60%) CTHRC1-correlated were ECM-related genes (Figure 2B).

Subsequently, an in vitro study was conducted; PSCs were treated with PBS, TGF-β (2.1 nM), or CTHRC1 (1.7 nM) (Figure 2C), and the genes that were not significantly changed by CTHRC1 treatment were noted (Appendix A), revealing that 15 ECM-related genes were upregulated. To confirm the regulatory role of CTHRC1, the mRNA expression of CTHRC1 was inhibited using siCTHRC1 or siCon (as a negative control), and the mRNA expression of five of the most upregulated genes (FAP, CTSK, LUM, POSTN, and SPARC) were then measured. The inhibition of CTHRC1 (by >90%) led to a 40–70% decrease in the genes in PSCs (Figure 2D).

### 3.3. CTHRC1 Activated PSCs In Vitro

After 24 h treatment with PBS, TGF-β (2.1 nM), or CTHRC1 (1.7 nM), PSC activation was evaluated [11] using the following procedure: (a) FAP and α-SMA protein expressions were determined using Western blot and IF assays; (b) collagen I and fibronectin concentrations in the PSC culture media were measured; (c) lipid droplets in the cytoplasm of PSCs were quantified using Oil Red O staining; and (d) the migratory/invasive capacity of PSCs were assessed using migration/invasion assays.

After the treatment with TGF-β or CTHRC1, PSCs showed typical signs of activation, including an increased expression of FAP and α-SMA (Figure 3A,B and Appendix A show uncropped images of Western blot), reduced optical density (O.D.) of Oil Red O staining (Figure 3C), and increased migration and invasion of PSCs (Figure 3D). The activating effects of CTHRC1 on PSCs were further confirmed using an anti-CTHRC1 antibody (69.4 nM). CTHRC1-induced collagen I and fibronectin concentrations (Figure 3E), and CTHRC1-induced migratory/invasive capacity of PSCs (Figure 3F) were all significantly blocked by the anti-CTHRC1 antibody, while the antibody (PBS + α-CTHRC1 group) did not show any suppressive effects compared to PBS alone.

In addition, the above studies all used immortalized PSCs (hPSC21-S/T). Key in vitro studies were conducted using both primary PSCs and immortalized PSCs to evaluate the potential differences in their responses to CTHRC1 because immortalized PSCs may differ considerably from primary PSCs in terms of their responses to pancreatic cancer cells [18]. Our results show no significant differences between the two types of PSCs in the responses to CTHRC1 (Appendix A).

### 3.4. CTHRC1 Differentiated PSCs into myCAFs In Vitro

The bioinformatic analyses of the TCGA data showed that the CTHRC1 clustered more closely to the myCAF marker genes (ACTA2, TAGLN, THY1, and POTN) than iCAF (CLEC3B, COL14A1, and IL-6) or apCAF (SLPI and CD74) marker genes (Figure 4A). CAF markers were defined by previous studies [19,20]. Treating PSCs with CTHRC1 (1.7 nM) led to a significant upregulation of myCAF marker genes only, but not iCAF or apCAF marker genes (Figure 4B). In contrast, adding the anti-CTHRC1 antibody (69.4 nM) only suppressed the gene expression of myCAFs but not iCAFs or apCAFs (Figure 4C). In terms of the PSC function, CTHRC1 treatment markedly increased the genetic expression of tumor-promoting growth factors (EGF, HGF, and FGF by 4.9, 4.4, and 10.5 times, respectively) and cytokines (IL8 and IL10 by 3.7 and 2.9 times, respectively) (Figure 4D).

### 3.5. CTHRC1 Promoted Pancreatic Cancer Growth via Activated PSCs

Pancreatic cancer proliferation was determined by WST-1 assays. In the first proliferation study, cancer cells were treated directly by PBS or CTHRC1 and also by conditioned media (Figure 5A). Compared to a direct culture of pancreatic cancer cells with PBS/CTHRC1, the treatment of pancreatic cancer cells with PSC-conditioned media obtained from PBS/CTHRC1-treated PSCs resulted in a significantly higher cancer cell proliferation. Additionally, CTHRC1-low-expressing MIA PaCa-2 cells showed a much higher CTHRC1-induced cell proliferation compared to the CTHRC1 high-expressing BxPC-3, under both direct culture (1.4 vs. 1.2 folds of Ctrl) and PSC CM (2.1 vs. 1.6 folds of Ctrl) conditions (Figure 5B).

In the second proliferation study, cancer cells were treated with conditioned media only (Figure 5C). PSC-mediated CTHRC1 cancer proliferation-promoting effects were completely reversed by the anti-CTHRC1 antibody in both cell lines (Figure 5D). To determine whether the proliferation-promoting effect of PSC CM was caused by additional CTHRC1 secretion from PSC, we measured CTHRC1 concentrations in PSC-conditioned media with or without treatment with CTHRC1, and no differences were found (Figure 5E). In addition, 24 and 72 h of PSC culture with CTHRC1 did not cause PSC proliferation (Figure 5F). Moreover, anti-CTHRC1 (up to 20 μg/mL) antibody was not cytotoxic to pancreatic cancer cells (Figure 5G).

### 3.6. Periostin Was Essential in the CTHRC1–PSCs–Cancer Metastasis Axis

POSTN is a myCAF biomarker [19,20] and our in vitro studies demonstrated that periostin (coded by the POSTN gene) expression in PSCs was highly regulated by CTHRC1 treatment (Figure 2C and Figure 4B,C). Therefore, we aimed to further clarify the relationship between periostin, pancreatic cancer cells, and PSCs, and to explore the molecular functions of periostin.

Periostin was not secreted by any of the four pancreatic cancer cell lines (MIA PaCa-2, PANC-1, CFPAC-1, and BxPC-3) with or without treatment with CTHRC1 (10, 100, or 1000 ng/mL). CTHRC1 secretion in the cancer cell lines was not influenced by periostin treatment either (Figure 6A). However, treatment with CTHRC1 led to a significantly increased protein expression of periostin in PSCs (Figure 6B and Appendix A shows uncropped images of Western blot), which showed clear CTHRC1 dose dependency (at 0, 10, 100, and 1000 ng/mL, R^2^ = 0.9676 by log-transformed linear regression, *p* < 0.001, Figure 6C). Moreover, the CTHRC1-induced periostin increase was significantly blocked by the anti-CTHRC1 antibody (Figure 6D).

The knockout of the POSTN gene in PSCs caused a very mild (but statistically significant) reduction in collagen I and fibronectin production, and very little change in the responsive ability of collagen I and fibronectin production to CTHRC1 (1.7 nM) stimulation (Figure 6E). However, the knockout of the POSTN gene caused the PSC-conditioned media to completely lose its cancer migration/invasion-promoting effects in response to the stimulation of CTHRC1 (Figure 6G). Notably, CTHRC1 was removed from the PSC-conditioned media (Appendix A) to exclude interference from CTHRC1 on cancer cell migration/invasion.

The high tumor mRNA expression of POSTN was associated with a significantly worse overall survival (all-time risk of death: HR = 2.3 and 3.8 for patients from the TCGA and CPTAC programs, respectively, Figure 6H) of pancreatic cancer patients. Furthermore, the incorporation of POSTN and CTHRC1 mRNA expression (POSTN^2^ × CTHRC1) led to a slightly improved prediction of overall survival (Figure 6I).

### 3.7. Inhibition of CTHRC1 Suppressed Tumor Growth and Induced Tumor myCAF Differentiation into iCAFs In Vivo

To investigate the influence of the anti-CTHRC1 antibody in vivo, two human pancreatic-cancer-bearing mouse models were established (subcutaneous BxPC-3, Figure 7A; and orthotopic CFPAC-1, Figure 7E). In both models, treatment with the anti-CTHRC1 antibody resulted in a significant tumor growth suppression; the combination of gemcitabine and the anti-CTHRC1 antibody caused the most significant tumor suppression in the orthotopic model (Figure 7B,F and Appendix A show tumor images). In the subcutaneous model, the histochemical staining of tumor tissue samples for Ki67 and collagen showed a significant decrease in Ki67 in mice treated with gemcitabine or the anti-CTHRC1 antibody (Figure 7C). However, there were no statistically significant differences between the two treatment groups (Figure 7D). In the orthotopic models, collagen staining was significantly reduced in the anti-CTHRC1 group only, irrespective of monotherapy or combination with gemcitabine (Figure 7G).

The tumor gene expression analyses showed that anti-CTHRC1 suppressed myCAF markers in tumor tissues; however, no suppression reached statistical significance. Unexpectedly, iCAF markers (CLEC3B and IL6) were largely and significantly upregulated in tumor tissues (Figure 8A). Additionally, the anti-CTHRC1 treatment led to a significant upregulation of factors with TME-suppression effects (GM-CSF, IL-1, and TGF-β), while a significant suppression of CTHRC1 was not noted (Figure 8B). These findings are different from the results of our in vitro study (Figure 4C).

## 4. Discussion

For the first time, this study demonstrated that pancreatic stellate cell (PSC) activation is a central mechanism of the pro-tumoral effects of CTHRC1 and elucidated the plasticity of the tumor microenvironment (TME) following CTHRC1-targeted pancreatic cancer treatment. Periostin, a stroma-specific molecule coded by the gene POSTN, was found to have a central role in the CTHRC1–PSC–cancer metastasis axis. The anti-CTHRC1 antibody was shown to reverse PSC activation and CAF differentiation, and to suppress tumor growth in animal models. However, long-term treatment may lead to an unexpected cancer-associated fibroblast (CAF) subtype shift from myofibroblast-like CAFs (myCAFs) to inflammatory CAFs (iCAFs).

We applied integrated bioinformatic in vitro and in vivo approaches to strengthen the scientific basis and clinical relevance of our results. The upregulation of CTHRC1 has been consistently recognized in multiple solid tumors [21]. Our analyses of the transcriptomes of pancreatic tumor and paired non-tumor tissue samples using publicly available Gene Expression Omnibus (GEO) datasets revealed very similar distribution patterns of CTHRC1 across multiple studies (Figure 1A). Moreover, the current study is the first to report that upregulated CTHRC1 in tumor tissue is most likely secreted by fibroblasts in cancer stroma, which are mainly stellate cells and CAFs. In pancreatic tumor tissue, cancer cells (ductal cells and/or acinar cells) account for only a minor part of all cellular components, and cancer stroma (consisting of PSCs, fibroblasts, endothelial cells, and various immune cells) makes up the remaining constituents [22,23]. Among the stromal cells, PSCs and fibroblasts are major sources of ECM [24].

We found that CTHRC1 was highly expressed in stromal cells. In PSCs, more than 60% of the genes that were highly correlated with CTHRC1 were shown to be ECM-related (including FAP, SULF1, and POSTN). Such correlations have been described in gastric cancer [25], where it was predicted that CTHRC1 may be heavily involved in common tumorigenic processes together with other molecules (through mechanisms such as co-expression, co-localization, and shared protein domain), although the exact function of CTHRC1 remained largely unknown at the time. Our in vitro studies clearly showed that CTHRC1 is a major regulatory factor of ECM-related genes when compared to well-established tumor-promoting cytokine TGF-β. The two molecules showed notable differences in regulating genes, such as THBS2, COL10A1, LUM, POSTN, and MFAP5, although in other genes, there were overlapping effects.

Previously, CTHRC1 was found to activate hepatic stellate cells (HSCs) and promote liver fibrosis [11]. The current study confirmed CTHRC1 induced PSC activation as well as exploring the fate of PSCs post-activation: the transformation of PSCs to CAFs, and more specifically, to myCAFs. A series of in vitro studies revealed that CTHRC1 and TGF-β showed a similar ability to activate PSCs at concentrations < 5 nM. These findings may be explained by the activation of common downstream molecules, such as Smad2/3 [11]. Notably, we found that CTHRC1 differentiated PSCs into myCAFs, as well as caused a surge in the upregulation of multiple growth factors and cytokines (Figure 4D). Such dramatic changes explain well the potent tumor proliferation-promoting effect of PSCs in response to CTHRC1 stimulation (Figure 5B). This suggests that the tumor-promoting effects of CTHRC1 is largely dependent on its stimulation of PSCs rather than on its autocrine effects. However, CTHRC1 did not stimulate PSCs to secrete CTHRC1 (contrasting its strong stimulation of other pro-tumoral molecules), nor did it promote the proliferation of PSCs (Figure 5E,F). This suggests that the regulatory effects of CTHRC1 on PSCs are likely imposed via influencing the functions of PSCs rather than increasing their proliferation. A previous study that conducted single-cell RNA-seq on pancreatic tumor tissue found that the numbers of CAFs (consisting of stellate cells and fibroblasts) do not differ significantly by disease stage [23]. Nevertheless, we found that neutralizing CTHRC1 completely reversed the activation of PSCs (migration/invasion/ECM secretion) and dramatically hindered the differentiation of PSCs to myCAFs, as well as blocked the pro-tumoral effects mediated by activated PSCs (e.g., cancer proliferation).

This study also presented important findings about the interactions between CTHRC1 and periostin. Periostin is an ECM molecule that is involved in multiple cellular functions, such as cell adhesion and cell cycle regulation (Online Mendelian Inheritance in Man (OMIM), entry: 608777). In this paper, we present two key findings. First, the interactions between CTHRC1 and periostin likely occurs exclusively in PSCs, not in pancreatic cancer cells. This is because, consistent with previous studies [26], periostin expression was extremely low in pancreatic cancer cells and was not influenced by CTHRC1, and periostin was also shown not to influence the CTHRC1 levels of cancer cells (Figure 5A). Second, the interactions between CTHRC1 and periostin may influence specific cellular functions only. This is evidenced by the knockout of the POSTN gene causing only a very mild decrease in the ECM secretion of PSC (Figure 6E) and the complete loss of its metatisis-promoting ability in response to CTHRC1 stimulation (Figure 6G).

The level of POSTN has been reported as a prognostic marker for multiple solid tumors [27]. Our analyses of the TCGA and CPTAC datasets showed that the higher expression of POSTN may be a greater predictor of mortality than CTHRC1. Interestingly, when we combined the two biomarkers (giving POSTN more weight) for survival prediction, the predictive performance was better (higher statistical significance) than using only CTHRC1 or POSTN alone. We speculate that, although CTHRC1 is an important multi-functional regulatory molecule in the TME, its regulatory effects on cancer cell migration/invasion are largely dependent on the POSTN expression of PSCs. Therefore, as a prognostic factor, POSTN had a higher HR than CTHRC1. However, the improved prediction ability conferred by the addition of CTHRC1 to POSTN may reflect the influences of other prognostic factors on survival. Nevertheless, it is important to note that these clinical datasets were not designed to study CTHRC1 or POSTN. Thus, our findings should be tested prospectively.

The influence of CTHRC1 was further tested in subcutaneous and orthotopic human pancreatic cancer mouse models to demonstrate proof of concept. The subcutaneous model is advantageous when it comes to measuring tumor growth, while the orthotopic model can better mimic the actual TME. In both models, anti-CTHRC1 antibodies significantly suppressed tumor growth to a similar or slightly less extent than gemcitabine. Although the anti-CTHRC1 antibody is known as a non-cytotoxic agent, the antibody clearly worked through reducing cancer proliferation and ECM formation. This was indicated by the significantly decreased Ki67 and collagen staining in tumor tissues in the anti-CTHRC1 group, compared to the control group in the subcutaneous model. In the orthotopic model, a combination of gemcitabine and anti-CTHRC1 produced the most significant tumor suppressive effects. This may be a complex result of gemcitabine’s cytotoxicity, and the anti-CTHRC1 antibody’s inhibition of tumor ECM production and facilitation of gemcitabine sensitivity through the inhibition of growth factors, such as HGF. Of note, previous studies consistently showed that interactions between PSC-derived HGF and cancer-cell-derived c-MET are the most important mechanism of pancreatic cancer resistance to gemcitabine [28] and targeting the HGF/c-MET axis is a key element in sensitizing the gemcitabine treatment of pancreatic cancer [29].

Animal studies were used to investigate the influence of long-term CTHRC1 treatment on PSC/CAFs that were otherwise unevaluable in vitro. In the in vitro study, CTHRC1 differentiated PSCs exclusively into myCAFs, and anti-CTHRC1 antibody completely reversed the differentiation. However, in the orthotopic model, we discovered the unexpected differentiation of myCAFs to iCAFs in tumor tissues after a long-term (4-week) treatment with the anti-CTHRC1 antibody. This is particularly important because it serves as strong evidence for the plasticity of the TME and helps to explain why the depletion of ECM production or inhibition of only a certain subtype of CAF has not yet improved patient outcomes in clinical trials [30].

This study has several limitations. First, because our experiments simultaneously investigated multiple molecules/markers, for simplicity’s sake, some in vitro studies were tested at the mRNA level only, rather than at the protein level. Second, previous studies showed that, in addition to PSCs, other pancreatic cancer stromal cells (such as endothelial cells and macrophages) can also differentiate into CAFs [24]. These stromal cells also secrete CTHRC1, as shown by the single-cell RNA-seq analysis (Figure 1D). Whether CTHRC1 can differentiate these cells into CAFs should be studied. Third, both animal models used in this study were human pancreatic cancer models using nude mice. It is unclear how different the interactions between mouse stromal cells and human cancer cells are from those between human stromal and cancer cells. Finally, although we found that anti-CTHRC1 treatments significantly suppressed tumor growth in the mice models, whether the shift of myCAFs to iCAFs has anti-tumor effects or influences survival should be studied.

## 5. Conclusions

This study showed that CTHRC1, a secreted protein that is highly expressed in pancreatic tumor tissue, has a critical role in ECM remodeling through PSC activation and CAF differentiation. CTHRC1 appears to be a potential therapeutic target for the treatment of pancreatic cancer, and the complexity and plasticity of the TME should be considered during the further development of anti-CTHRC1 treatments.

## 6. Patents

The anti-CTHRC1 antibody used in this study was registered as a patent in the following countries (regions): Republic of Korea (KR102487687B1, 2023) and South Africa (Patent No. not issued, 2023). The patent application is pending in another 21 countries (regions).

## Figures and Tables

**Figure 1 cancers-15-03370-f001:**
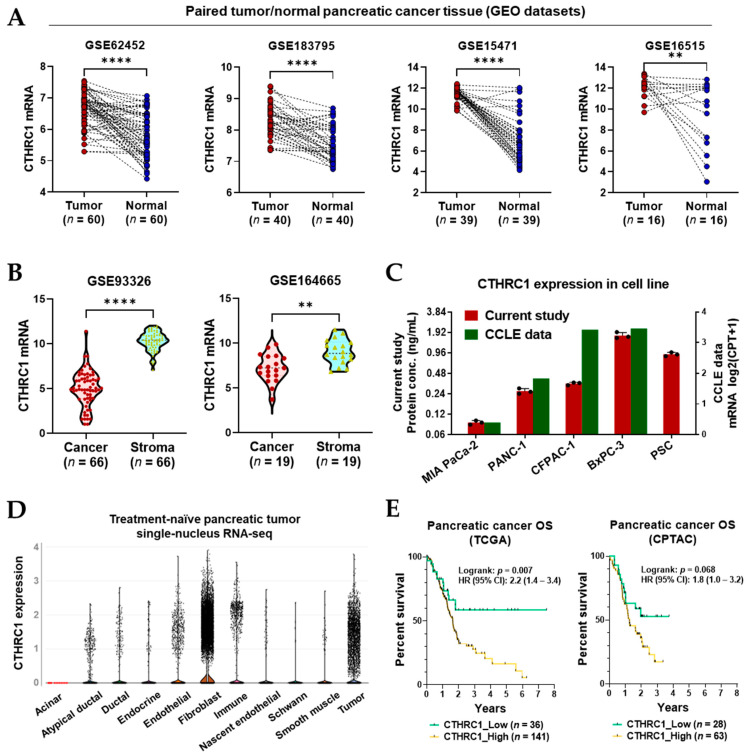
CTHRC1 is mainly expressed in pancreatic tumor stroma and a higher expression predicts poorer survival. CTHRC1 expression comparison based on datasets retrieved from the Gene Expression Omnibus (GEO) (**A**) in pancreatic tumor vs. paired adjacent normal tissue, and (**B**) in tumor stroma vs. cancer (epithelium) cells. (**C**) Expression of CTHRC1 in cancer cell lines and PSCs. Data were obtained from the current study, which measured CTHRC1 concentration, as well as from the CCLE database, which reported CTHRC1 mRNA expression levels. (**D**) CTHRC1 gene expression in different subsets of cells isolated from treatment-naïve pancreatic tumor tissue, determined by single-nucleus RNA-seq, retrieved from the Single-Cell Portal. (**E**) Survival analyses based on the CTHRC1 gene expression of tumor samples in pancreatic cancer patients from the TCGA (*n* = 177) and CPTAC (*n* = 91) programs. The low and high expression of CTHRC1 were divided when the highest statistical difference was reached. *p*-values: ** and **** indicate *p* < 0.05 and 0.0001, respectively.

**Figure 2 cancers-15-03370-f002:**
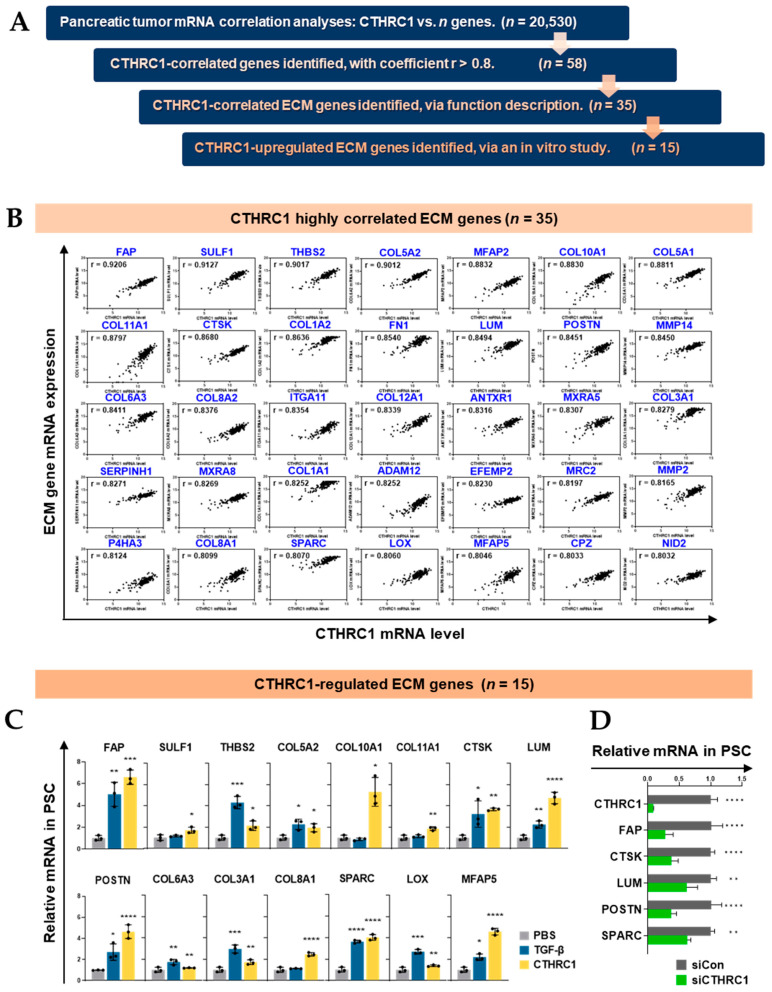
CTHRC1 regulated a considerable portion of extracellular matrix (ECM) genes. (**A**) The process of identification of ECM genes upregulated by CTHRC1. Based on the mRNA expression of pancreatic cancer tissue samples (*n* = 183) obtained from the TCGA PanCancer Atlas, 20,530 genes were screened for correlation with CTHRC1, and 58 genes showed a high correlation with CTHRC1 (r > 0.8). Among these 58 genes, (**B**) 35 were ECM-related genes. (**C**) In vitro, pancreatic stellate cells (PSCs; in this study, it refers to immortalized PSC hPSC21-S/T) were treated with PBS, TGF-β (2.1 nM), or CTHRC1 (1.7 nM) for 24 h, and 15 of the 35 ECM-related genes showed significant upregulation by the CTHRC1 treatment. (**D**) Relative mRNA expression of five of the most CTHRC1-upregulated genes in PSCs after inhibition by siCTHRC1 vs. siCon (control). *p*-values: *, **, *** and **** indicate *p* < 0.05, 0.01, 0.001 and 0.0001, respectively.

**Figure 3 cancers-15-03370-f003:**
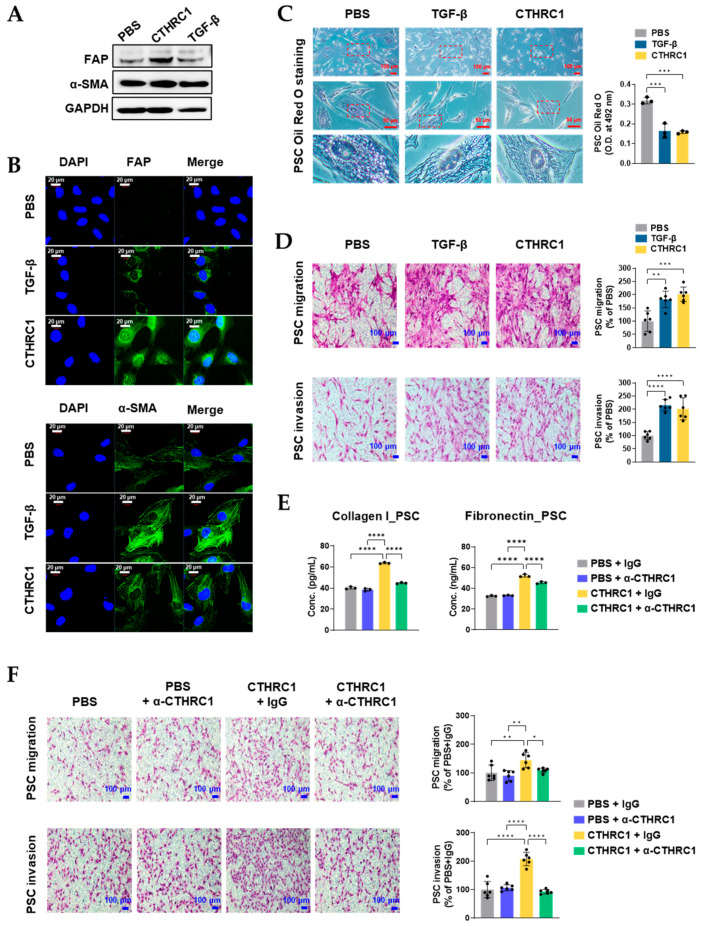
CTHRC1 activated pancreatic stellate cells (PSCs) in vitro. Activation of PSC after treatment with PBS, CTHRC1 (1.7 nM), or TGF-β (2.1 nM) for 24 h demonstrated by the protein expression of FAP and α-SMA in PSCs using (**A**) Western blot and (**B**) immunofluorescence (IF) assay, by (**C**) quantification of lipid droplets in PSC cytoplasm using Oil Red O staining (decreased optical density (O.D.) of lipid droplets indicates PSC activation), and by (**D**) metastatic capacity of PSCs using migration/invasion assays. To confirm the activating effects of CTHRC1 on PSCs, PSCs were treated with PBS, CTHRC1 (1.7 nM), and/or α-CTHRC1 (anti-CTHRC1 mAb, 69.4 nM) for 24 h, (**E**) concentrations of collagen I and fibronectin were measured using indirect ELISA, and (**F**) metastatic capacity of PSCs was determined using migration/invasion assays. *p*-values: *, **, *** and **** indicate *p* < 0.05, 0.01, 0.001 and 0.0001, respectively.

**Figure 4 cancers-15-03370-f004:**
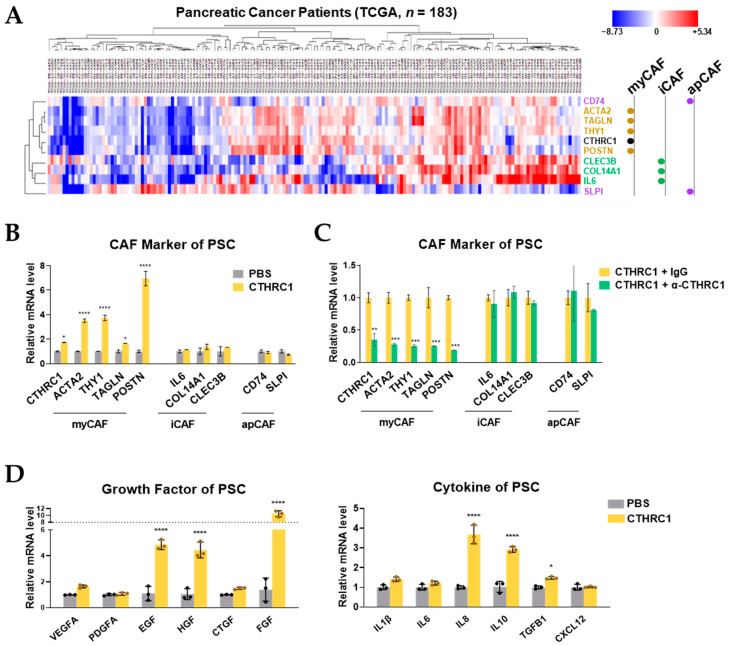
CTHRC1 differentiated pancreatic stellate cells (PSCs) into myofibroblast-like cancer-associated fibroblasts (myCAFs) in vitro. (**A**) Clustering of the mRNA expression of marker genes of myCAFs (ACTA2, TAGLN, THY1, and POSTN), inflammatory iCAFs (CLEC3B, COL14A1, and IL6), and antigen-presenting apCAFs (CD74 and SLPI) in pancreatic cancer tissue samples (*n* = 183) obtained from the TCGA PanCancer Atlas. Relative mRNA expression of CAF marker genes in PSCs (**B**) after treatment with PBS or CTHRC1 (1.7 nM) and (**C**) after treatment with CTHRC1 + IgG or CTHRC1 + α-CTHRC1 (anti-CTHRC1 mAb, 69.4 nM). (**D**) Relative mRNA expression of growth factors and cytokines in PSCs after treatment with PBS or CTHRC1. *p*-values: *, **, *** and **** indicate *p* < 0.05, 0.01, 0.001 and 0.0001, respectively.

**Figure 5 cancers-15-03370-f005:**
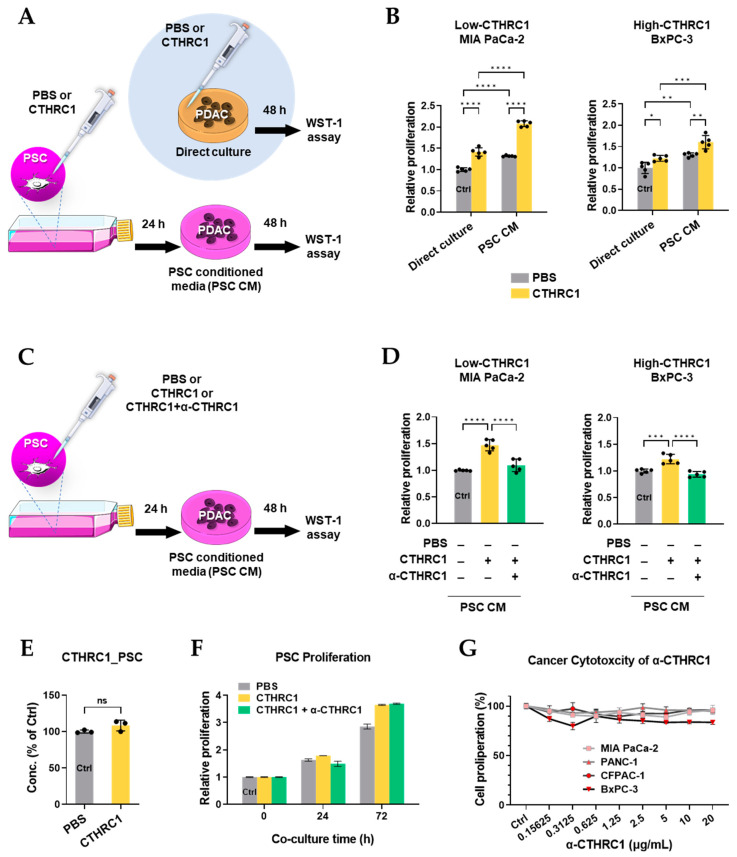
CTHRC1 promoted pancreatic cancer proliferation via pancreatic stellate cell (PSC) activation more than via direct stimulation. Design of the 1st cancer proliferation experiment shown in (**A**) cancer cells treated directly by PBS or CTHRC1 (1.7 nM) for 48 h, or by conditioned media (CM) harvested from the supernatant of PSCs cultured with PBS or CTHRC1 (1.7 nM) for 48 h, and (**B**) cancer proliferation measured in CTHRC1-low-expressing MIA PaCa-2 vs. CTHRC1-high-expressing BxPC-3 cells after PSC CM treatment vs. direct culture. Design of 2nd cancer proliferation experiment shown in (**C**) cancer cells treated with CM harvested from the supernatant of PSC cultured with PBS, or CTHRC1 (1.7 nM) with or without α-CTHRC1 (anti-CTHRC1 mAb, 69.4 nM) for 48 h. (**D**) Cancer proliferation measured in CTHRC1-low-expressing MIA PaCa-2 vs. CTHRC1-high-expressing BxPC-3 cells, following treatment with CTHRC1 PSC CM. (**E**) CTHRC1 concentrations measured in PSC supernatant harvested with or without treatment with CTHRC1 (1.7 nM) for 24 h. (**F**) The influence of CTHRC1 and CTHRC1 + α-CTHRC1 on 24 and 72 h PSC proliferation. (**G**) Cytotoxicity of CTHRC1 (0–20 μg/mL) in pancreatic cancer cell lines. *p*-values: *, **, *** and **** indicate *p* < 0.05, 0.01, 0.001 and 0.0001, respectively.

**Figure 6 cancers-15-03370-f006:**
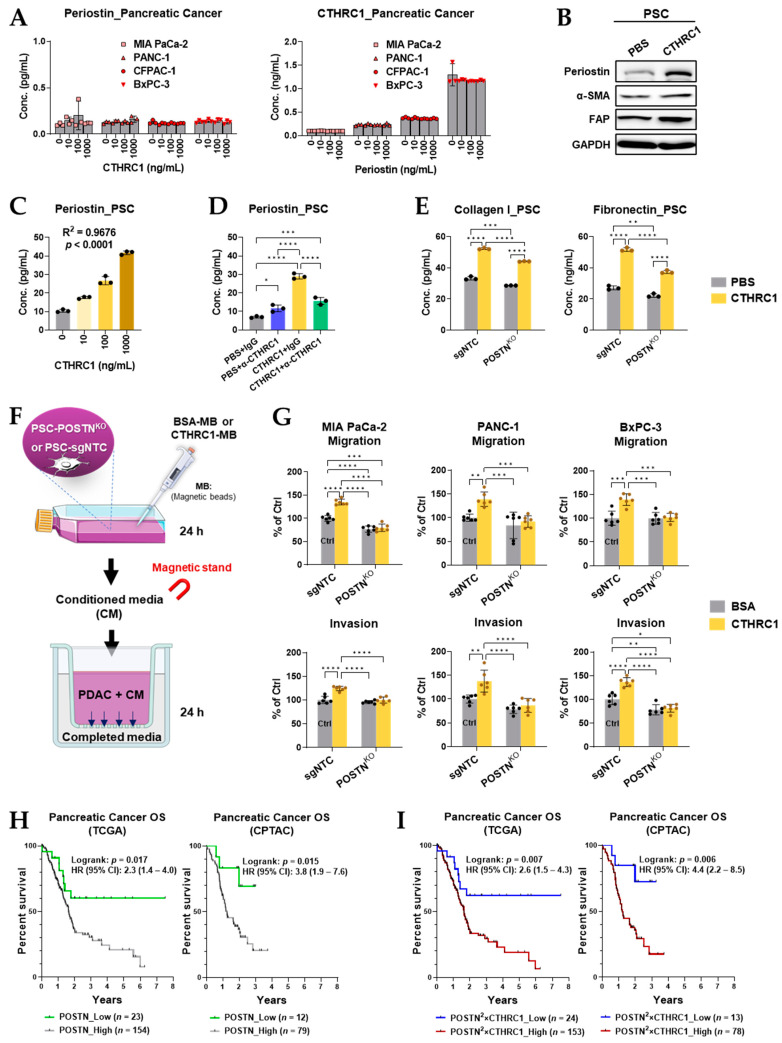
Periostin (coded by the POSTN gene), a molecule that interacts with CTHRC1 specifically via pancreatic stellate cells (PSCs), not pancreatic cancer cells, had an essential role in the CTHRC1–PSC–cancer metastasis axis. (**A**) Concentrations of periostin and CTHRC1 in the pancreatic cancer cell supernatant after treatment with CTHRC1 and periostin (0, 10, 100, and 1000 ng/mL), respectively, for 24 h. (**B**) Protein expression of periostin in PSCs after treatment with PBS or CTHRC1 (1.7 nM) for 24 h. Concentration of periostin in PSCs (**C**) after treatment with 0, 10, 100, and 1000 ng/mL of CTHRC1, and (**D**) after treatment with PBS/CTHRC1 (1.7 nM) + IgG vs. PBS/CTHRC1 + α-CTHRC1 (anti-CTHRC1 mAb, 69.4 nM). (**E**) Concentrations of collagen I and fibronectin in PSCs with POSTN-knockout (PSC_POSTN^KO^) after treatment with PBS or CTHRC1 (1.7 nM) for 24 h. (**F**) Diagrams of experiments designed to investigate the influences of the POSTN-knockout of PSCs (PSC-POSTN^KO^) on pancreatic cancer migration/invasion. (**G**) Migration and invasion of pancreatic cancer cells compared 24 h after adding conditioned media (CM), harvested from CTHRC1-MB (magnetic beads)-treated or BSA-MB-treated PSC_sgNTC or PSC_POSTN^KO^. CTHRC1 removed from CM using a magnetic stand to exclude CTHRC1 interference. In pancreatic cancer patients from the TCGA (*n* = 177) and CPTAC (*n* = 91) programs, the overall survival (OS) data were compared between the (**H**) low and high tumor mRNA expression of POSTN, and the (**I**) low and high mRNA expression of POSTN^2^ × CTHRC1. The low and high expression groups of POSTN and CTHRC1 were divided when the highest statistical difference was reached. *p*-values: *, **, *** and **** indicate *p* < 0.05, 0.01, 0.001 and 0.0001, respectively.

**Figure 7 cancers-15-03370-f007:**
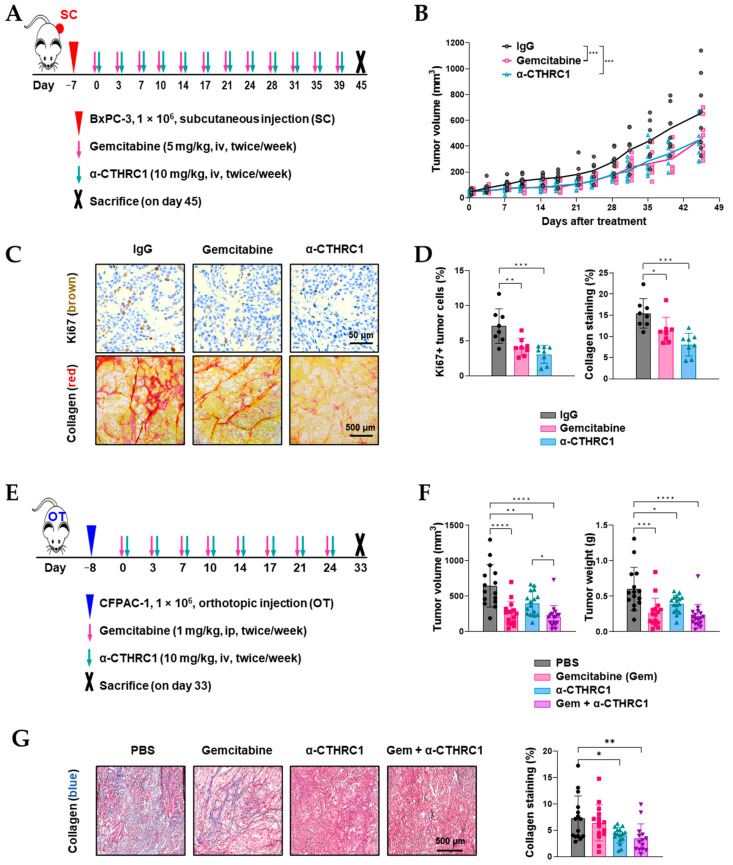
Anti-CTHRC1 antibody inhibited tumor growth and extracellular matrix (ECM) formation in vivo. (**A**) Diagram of experiment design in BxPC-3 subcutaneous model. (**B**) Tumor volume measured from day 0 to 45 in each animal group treated with PBS, gemcitabine, or α-CTHRC1 (anti-CTHRC1 mAb). (**C**) Representative images of the immunohistochemical staining of Ki67 and Sirius red staining of collagen in BxPC-3 tumor tissue and (**D**) quantification of Ki67-positive tumor cells and collagen staining. (**E**) Diagram of experiment design in CFPAC-1 orthotopic model. (**F**) Tumor volume and tumor weight of harvested CFPAC-1 tumors. (**G**) Representative images and quantification of Masson’s Trichrome staining of collagen in CFPAC-1 tumor tissues. *p*-values: *, **, *** and **** indicate *p* < 0.05, 0.01, 0.001 and 0.0001, respectively.

**Figure 8 cancers-15-03370-f008:**
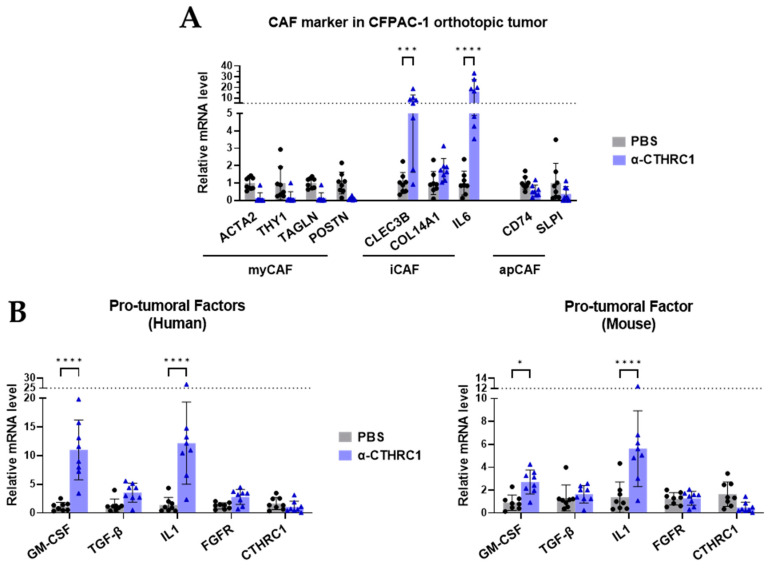
Long-term (4-week) anti-CTHRC1 treatment differentiated myofibroblast-like cancer-associated fibroblasts (myCAFs) into inflammatory iCAFs in the CFPAC-1 orthotopic mouse model. (**A**) Relative mRNA level of CAF marker genes in tumor tissue samples harvested from CFPAC-1 orthotopic mouse model, and (**B**) mRNA levels of pro-tumoral factors in CFPAC-1 tumor tissue detected using both human and mouse primers, compared between PBS vs. α-CTHRC1 (anti-CTHRC1 monoclonal antibody) treatment groups. *p*-values: *, *** and **** indicate *p* < 0.05, 0.001 and 0.0001, respectively. Solid circles●: PBS treatment, solid triangle▲: α-CTHRC1 treatment.

## Data Availability

Data are available upon request.

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
