# Peer review of "CTHRC1 Induces Pancreatic Stellate Cells (PSCs) into Myofibroblast-like Cancer-Associated Fibroblasts (myCAFs)"

_cancers, 2023, doi:10.3390/cancers15133370_

Round 1

Reviewer 1 Report

The paper by Min Kyung Kang et al., entitled “CTHRC1 induces pancreatic stellate cells (PSC) into myofibro-2 blast-like cancer-associated fibroblast (myCAF)” regards the Collagen triple helix repeat-containing protein 1, with recognized roles in wound healing, vascular remodeling and migration promotion  in cancer.

This work concerns the effects and expression levels of CTHRC1 in various cells, including immortalized pancreatic stellate cells (PSC). An initial informatic analysis based on Gene Expression Omnibus, a repository of high throughput gene expression and hybridization data or from the CCLE,  the cBioPortal, and TCGA PanCancer Atlas databases led to the conclusion that CTHRC1 is expressed in cancer stroma cells and expression is highly correlated with ECM-related molecules. Then, PSC are analysed after being exposed to soluble purified CTHRC1 to search for changes in gene expression. The responsive genes turned to be ECM-related and upregulated by CTHRC1 in PSC. Further work led to the conclusion that CTHRC1 can activate PSC, leading to the secretion of pro-tumoral components for prostate cancer cells. This is confirmed in vivo, assessing tumor growth inhibition by anti-CTHRC1 antibodies.

The use of a single immortalized pancreatic stellate cell line (PSC hPSC21-S/T, line) does not allow to extend the conclusions to all pancreatic cells. It is reported that commonly used cells are heterogenous in their response to pancreatic cells (Lenggenhager et al. Cells, 2019, 8, 1, 23). Therefore, at least some of the relevant experiments should be repeated on other immortalized PSC. 

The paper is not written in a clear manner, it is often ambiguous and cryptic, with many grammar mistakes and should definitely be re-written. Figure legends contain too many conclusions, that should be transferred to the Results section.

More comments follow:

Lines 289-290: “Four studies (GSE62452, GSE183795, GSE15471, and GSE16515) met the inclusion criteria and were used for the comparison of CTHRC1 expression between pancreatic tumors 290 and paired adjacent normal tissues” the paragraph should be introduced with the aim/strategy of the study and it is not clear whether the authors are citing any previous study or are referring to their first set of data.

Only figure 1 legend makes it clear that data are retrieved from the Gene Expression Omnibus (GEO), or from the CCLE database or from the cBioPortal in section A,B,D whereas C and E describe experimental data from the authors. This should be made more clear in the Results section.

Figure 2

A and B sections concern the informatic analysis based on TCGA PanCancer Atlas, whereas section C shows experimental results describing ECM-related genes upregulated by CTHRC1 in pancreatic stellate cells exposed to soluble purified CTHRC1 for 24 hours. Does this cell line corresponds to the one reported in Methods as the immortal human PSC line, hPSC21-S/T ?

In section D, mRNA expression of CTHRC1-upregulated genes is studied in PSC transfected with CTHRC1 siRNA. However, the two conditions are very different: in the latter case, the authors are affecting the level of endogenous CTHRC1. Is this acting in an autocrine fashion? Do they have evidence in that respect?

Figure 3 

Line 447: The effects of CTHRC1 on PSC, the authors make use of CTHRC1, and anti-CTHRC1 mAb. However, an important control to gain mechanistic insights would be to check whether cells treated with soluble CTHRC1 express β1 integrin, as Chen et al. reported that CTHRC1 promotes cell adhesion to ECM through β1 integrin and FAK activation (Chen et al., PLoS One, 2013, 8,7). 

Figure 5

Line 364-on: it is not explained whether the conditioned media of pancreatic stellate cells are indeed prepared in the absence of FBS and neither discussed if the conditioned media still contain CTHRC1 protein and the relative concentration. These details should be provided to evaluate the net effects of the secreted products by pancreatic stellate cells affecting pancreatic cancer cell lines (BxPC-3, MIA PaCa-2).

Lines 381-383

The rationale to study periostin is not explained, nor the need of assessing the level of periostin in pancreatic cancer cell lines (MIA PaCa-381 2, PANC-1, CFPAC-1, and BxPC-3), with or without exposure to CTHRC1. Which is the working model?

With this premise, it is impossibile to interpret the results in mechanistic terms, i.e CTHRC1- induction of periostin protein expression or gene knock-out-dependent reduction in collagen I and fibronectin.

Minor comment:

Line 293: “the information of the datasets was 293 presented in Supplementary Table 2.” Should be substituted with “the information of the datasets is presented in Supplementary Table 2.”

The paper is not written in a clear manner, it is often ambiguous and cryptic, with many grammar mistakes and should definitely be re-written. 

Author Response

Dear Reviewer,

Thank you so much for spending time reviewing our manuscript. We have responded to your comments point by point, please see the attached. In addition, we have sent our manuscript to a professional editor for English editing. Your comments have been very valuable to us and helped improve the article significantly. We hope the article now is suitable for publication now.

However, we are more than willing to do more revisions if you have other comments.

Best regards
Fen Jiang

Reviewer 2 Report

Kang et. al investigated the role of CTHRC1 in ECM remodeling through PSC activation and CAF differentiation. Blocking CTHRC1 reverses this process and exerts anticancer effects confirmed in in vivo studies and may be a potential therapeutic target. This model of anti-cancer therapy requires further study as iCAF formation has been observed following long-term inhibition of CTHRC1. The research sheds new light on the pro-tumoral effects of CTHRC1 through activation of PSC.

Studies have been carried out in vitro and in vivo models. The authors used the original anti-CTHRC1 humanized monoclonal antibody (mAb). The antibody is patented, but it would be good if the authors provided more details in the publication about the nature of the antibody and its production.

Author Response

Dear Reviewer,

Thank you so much for spending time reviewing our manuscript. 

We appreciate your comment "It would be good if the authors provided more details in the publication about the nature of the antibody and its production."

Currently we are preparing another manuscript to introduce the engineering and characterization of the patented anti-CTHRC1 antibody.

However,  I have attached the certificate of the patent below in case you may find it interesting to read:

https://patents.google.com/patent/KR102487687B1/en

In addition, we have made considerable changes based on other reviewers' comments as well, we hope the manuscript is much improved for publication now.

Best regards
Fen Jiang

Reviewer 3 Report

Comments to the corresponding author: Supplementary materials weren’t available as the cited link doesn’t work.

Author Response

Dear Reviewer,

Thank you so much for spending time reviewing our manuscript. And your positive feedback has been a great encouragement to us. 

As to your comment "Supplementary materials weren’t available as the cited link doesn’t work." We have attached all the supplemenatry materials.

In addition, we have made considerable changes to the manuscript based on other reviewer's comments, and we hope the article is much improved now for publication.

Best regards
Fen Jiang
